# GLUT1 Deficiency Syndrome—Early Treatment Maintains Cognitive Development? (Literature Review and Case Report)

**DOI:** 10.3390/genes12091379

**Published:** 2021-08-31

**Authors:** Ivana Kolic, Jelena Radic Nisevic, Inge Vlasic Cicvaric, Ivona Butorac Ahel, Kristina Lah Tomulic, Silvije Segulja, Kristina Baraba Dekanic, Senada Serifi, Aleksandar Ovuka, Igor Prpic

**Affiliations:** 1Pediatric Clinic, Clinical Hospital Center Rijeka, Kresimirova 42, 51000 Rijeka, Croatia; jelaradi@yahoo.com (J.R.N.); ivonabuah@gmail.com (I.B.A.); klahtomulic@gmail.com (K.L.T.); silvijesegulja@gmail.com (S.S.); k.baraba.dekanic@gmail.com (K.B.D.); senada.serifi@gmail.com (S.S.); aovuka1@gmail.com (A.O.); 2Department of Clinical Science, Faculty of Health Studies, University of Rijeka, Viktora Cara Emina 5, 51000 Rijeka, Croatia; 3Department of Pediatrics, School of Medicine Rijeka, University of Rijeka, Brace Branchetta 20/1, 51000 Rijeka, Croatia; inge_v_c@yahoo.com; 4Department of Clinical, Health and Organizational Psychology, Clinical Hospital Center Rijeka, Kresimirova 42, 51000 Rijeka, Croatia

**Keywords:** early diagnosis, genotype, ketogenic diet, myoclonic epilepsy, phenotype

## Abstract

Glucose transporter type 1 (GLUT1) is the most important energy carrier of the brain across the blood–brain barrier, and a genetic defect of GLUT1 is known as GLUT1 deficiency syndrome (GLUT1DS). It is characterized by early infantile seizures, developmental delay, microcephaly, ataxia, and various paroxysmal neurological phenomena. In most cases, GLUT1DS is caused by heterozygous single-nucleotide variants (SNVs) in the *SLC2A1* gene that provoke complete or severe impairment of the functionality and/or expression of GLUT1 in the brain. Despite the rarity of these diseases, GLUT1DS is of high clinical interest since a very effective therapy, the ketogenic diet, can improve or reverse symptoms, especially if it is started as early as possible. We present a clinical phenotype, biochemical analysis, electroencephalographic and neuropsychological features of an 11-month-old boy with myoclonic seizures, hypogammaglobulinemia, and mildly impaired gross motor development. Using sequence analysis and deletion/duplication testing, deletion of an entire coding sequence in the *SLC2A1* gene was detected. Early introduction of a modified Atkins diet maintained a seizure-free period without antiseizure medications and normal cognitive development in the follow-up period. Our report summarizes the clinical features of GLUT1 syndromes and discusses the importance of early identification and molecular confirmation of GLUT1DS as a treatable metabolic disorder.

## 1. Introduction

Glucose is an essential fuel for brain energy metabolism [1]. Glucose transport across the blood–brain barrier (BBB) and the astrocyte plasma membrane is exclusively facilitated by glucose transporter type 1 (GLUT1). This membrane-bound protein is encoded by *SLC2A1*, a gene on chromosome 1p35-31.3 (OMIM 138140) [1,2,3].

Glucose transporter type 1 deficiency syndrome (GLUT1DS, OMIM 606777) is a rare neurological disorder caused by impaired glucose delivery to the brain. It results from haploinsufficiency of the *SLC2A1* gene [4]. It was first described by De Vivo et al. in 1991, who reported two patients with a novel clinical syndrome characterized by an infantile-onset epileptic encephalopathy associated with delayed neurological development, deceleration of head growth, acquired microcephaly, incoordination, and spasticity [5].

Patients classically present with infantile-onset epilepsy, impaired neurological development, complex movement disorders, intellectual disability, and acquired microcephaly. Several atypical variants of GLUT1DS have also been recognized [5,6,7].

Diagnosis is confirmed by the evidence of hypoglycorrhachia and loss-of-function variants in the *SLC2A1* gene [4]. Most cases are sporadic; only rare families with *SLC2A1* variants have been described with an autosomal dominant mode of inheritance [2].

The treatment of choice for GLUT1DS is a high-fat, low-carbohydrate diet that mimics the metabolic state of fasting. This ketogenic diet provides ketones as an alternative fuel for the brain and effectively restores brain energy metabolism [6].

Our report summarizes the clinical features of GLUT1DS and discusses the importance of the early identification and molecular confirmation of GLUT1DS as a treatable metabolic disorder.

## 2. Materials and Methods

### 2.1. Acquisition of Clinical Case

A comprehensive evaluation of our patient was undertaken, including molecular genetic investigations. Informed consent in agreement with the Declaration of Helsinki was signed by the parents of the patient.

### 2.2. Developmental Test

The developmental profiles were based on scores achieved on the developmental test Čuturić for preschool children and Vineland scale.

The developmental test Čuturić is designed for testing the psychomotor development of infants, toddlers, and preschool children in the Croatian language. It is applicable to healthy children and children with developmental impairments. It is made of two parts. The first part is for children younger than 2 years of age, with 15 subscales of 10 tasks each. The second part is for children at 2 to 8 years of age, with 7 subscales of 6 tasks each.

It is performed by a trained clinical psychologist, and it tests visuoconstructive, visuoperceptive, graphomotor, speech, and motor development. It also tests counting, memory, writing, and reading skills and knowledge expression. 

The social maturity level was assessed using the Vineland-II Social Maturity Scale.

### 2.3. Modified Atkins Diet Protocol

A modified Atkins diet (MAD) is prescribed for infants, toddlers, and children with drug-resistant epilepsy and GLUT1DS in our clinic. The earliest age for starting the diet is 6 months. Introduction of the diet requires hospitalization of the patient. All children undergo a detailed clinical examination and baseline laboratory test prior to the start of the diet. A detailed clinical examination includes the nutritional status of the child (body weight, body height, and body mass index) and routine abdominal ultrasound. Baseline laboratory tests include complete blood count, analysis of kidney and liver function, and lipid testing. A diet plan is individualized, and before starting it, a clinical nutritionist provides education to parents. Daily energy requirements are divided in four to five meals according to the clinical nutritionist’s guidelines. Initial carbohydrate intake is between 10 and 15 g/day and is gradually raised to a maximum of 20 g/day. KetoCal^®^ is used in infants and toddlers as a very-high-fat, low-carbohydrate, and nutritionally complete liquid product, as a meal alone or mixed with other macronutrients. During the first month of diet, parents regularly measure the levels of glucose and ketones in the blood (twice a week), and they also write a diary that contains information on every meal, potential adverse effects of the diet, and laboratory test results. After the introduction of the diet, regular outpatient clinic visits are scheduled: every month for the first 3 months and then every 3 months. These visits include detailed clinical examination and laboratory results as mentioned above. If any adverse effects are noticed, the diet is discontinued.

## 3. Results

### 3.1. Case Presentation

An 11-month-old male child with a history of myoclonic jerks was referred to a pediatric neurologist. He was born at term out of an uneventful pregnancy and delivery as the first child of healthy nonconsanguineous parents. The family history was unremarkable. The birth weight, length, and head circumference were according to gestational age (2870 g, 51 cm, and 35 cm, respectively). At the age of 4 months, the parents noticed jerking of the hands and legs in light sleep, and at the age of 6 months, jerking was noticed in awake state and during normal activities. At the age of 8 months, head nodding was noticed, emphasized several weeks prior to admission. 

The clinical examination at admission was unremarkable.

The first developmental test at the age of 12 months revealed impaired gross motor development (developmental age of 10 months according to tests) and slightly impaired hand function (no pincer grasp developed). Language development was in the preverbal phase, according to age, and his emotional and social skills were normal. 

The electroencephalographic (EEG) study at admission registered an ictal EEG during head myoclonus with a short generalized spike-wave discharge, followed by diffuse slowing, and an interictal EEG was normal (normal background activity without slowing and epileptiform discharges, respectively). Levetiracetam treatment was started, and diagnostic workup was continued.

Laboratory findings revealed low levels of immunoglobulins A, G, and M (<0.2 g/L, 0.5 g/L, 1.2 g/L, respectively), while others were normal (including metabolic workup). Magnetic resonance imaging of the brain was normal. Analysis of cerebrospinal fluid (CSF) in fasting state revealed hypoglycorrhachia (2.0 mmol/L), and the CSF-to-blood-glucose ratio was 0.4 with normal lactate levels (0.9 mmol/L).

History, seizure semiology, and detected hypoglycorrhachia raised suspicion in GLUT1DS, so a modified Atkins diet was started promptly, and molecular genetic testing was scheduled. Several days after starting the diet, the child was seizure free, and levetiracetam was discontinued after 2 months. 

Now, at the age of 2 years, our patient is seizure free without complications of the diet, and according to developmental tests, his motor, verbal, and social skills are normal for his age. His head circumference is normal for his age. Hypogammaglobulinemia is still present without clinically severe infections.

### 3.2. Molecular Genetic Analysis

Due to the phenotypic presentation, we confirmed the suspected diagnosis of GLUT1DS. NGS identified a full coding sequence deletion of the *SLC2A1* (NM_006516.2) gene. Sanger sequence analysis confirmed the presence of the heterozygous variant in the proband, and the healthy parents were negative. The variant was classified as pathogenic according to the ACMG guidelines [8] and the consistent clinical phenotype.

## 4. Discussion

GLUT1 deficiency syndrome is a treatable metabolic disorder affecting the nervous system caused by poor glucose transport at the cerebral level and clinically characterized by a variety of neurological signs and symptoms [9]. The classic GLUT1DS phenotype is a metabolic encephalopathy comprising a range of complex movement disorders, epilepsy, mental retardation, deceleration of head growth, and acquired microcephaly, with onset below 1 year of age [10]. Symptoms develop in an age-specific pattern: paroxysmal eye–head movements and seizures are early presenting features in infancy. Developmental impairment becomes increasingly apparent and is followed by ataxia, paroxysmal-exertion-induced dystonia, and further movement abnormalities that develop over time, often becoming the major symptoms in adolescents and adult GLUT1DS patients [7]. Clinical severity varies from mild motor and cognitive dysfunction between epileptic attacks to severe neurological disability, with some patients never achieving language or unsupported walking [10].

When considering previously reported clinical phenotypes, our patient can be categorized into a group of classical phenotypes with myoclonic seizures and head nodding starting at infancy with normal head growth and intelligence for his age. Spinal fluid analysis in our patient was according to literature reports, hypoglycorrhachia, low CSF-to-glucose ratio, and low-normal lactate levels.

Since the initial description of GLUT1DS in 1991, the number of affected individuals has steadily grown, facilitated by the advance of molecular diagnosis [7]. Sequence analysis identifies heterozygous pathogenic variants (or rarely, biallelic pathogenic variants) in *SLC2A1* in 81–89% of patients [7]. The majority of patients carry de novo variants [11]; only rare families with *SLC2A1* mutations have been described with an autosomal dominant mode of inheritance [2]. Most variants found in *SLC2A1* are heterozygous missense mutations with an autosomal dominant effect. However, frameshift mutation; deletions; insertions; and intronic, promotor, and splice-site mutations have also been described [2]. Generally, patients with missense mutations often present with moderate to mild symptoms, but the genotype–phenotype correlation in GLUT1DS is complex, and it is not yet clearly defined. However, it has been speculated that large-scale deletions and nonsense, frameshift, and splice-site mutations result in 50% loss of the GLUT1 protein and are associated with the moderate classical phenotype of GLUT1DS [10].

In our patient, gross deletion of the genomic region encompassing the full coding sequence of the *SLC2A1* gene was identified, and similar deletion has been rarely reported in individuals with GLUT1DS (PMID: 20382060) [12,13]. It has been previously described that patients with gross deletion in the *SLC2A1* gene present with a classical infant onset of GLUT1DS [13]. A possible explanation of the “milder” phenotype in our patient with large deletion in the *SLC2A1* gene could be confirmation at a very early stage of disease and early treatment, but without further research, we cannot make a definite conclusion on a genotype–phenotype correlation in our patient.

Cognitive deficits are common and may be severe in patients with classic GLUT1DS [14,15,16]. In milder forms, cognitive development may be less affected and, sometimes, even normal. Children with normal psychomotor development and adults with normal cognitive function have rarely been reported [13]. Regarding cognitive impairment in GLUT1DS patients, it has been found that patients with myoclonic seizures have a higher risk of cognitive impairment [17] and that mental disability is more severe in patients with earlier disease onset [13,15]. Besides psychomotor retardation, attention deficit and behavioral disorders are frequently associated [18,19]. In addition to gait disturbances (ataxic or spastic gait or both), speech disorder is frequent, mostly in the form of mixed dysarthria, which may include features of spastic, ataxic, and hyperkinetic dysarthria, and may interfere with speech intelligibility [19]. Cognitive impairment is not detected in our patient for now, but further regular psychological follow-up is needed for the detection of impairment of cognition or speech or behavioral disorder.

In GLUT1DS in patients of all ages, an interictal EEG is often normal. Abnormalities appear more common at certain ages: in infants, focal slowing and focal epileptiform discharges are more prevalent, whereas in children aged 2 years or older, a 2.5 to 4 Hz generalized spike-wave pattern is observed. An intriguing feature, when present, is preprandial EEG abnormality that improves with feeding as glucose is restored to a starving brain [7,20,21]. We recorded an ictal EEG during myoclonic seizure in our patient, and an interictal EEG was normal, as previously described.

There are many studies investigating the correlation between GLUT1 deficiency and immunological pathways and diseases. Two mostly investigated are those in systemic lupus erythematosus and rheumatoid arthritis [22,23]. Metabolic pathways must be strictly regulated to allow normal proliferation and T cell effector function [24,25,26]. GLUT1 is also essential for B cell homeostasis and antibody production. B cells require GLUT1 and glucose uptake to maintain B cell populations and to support metabolic reprogramming necessary for maximal antibody secretion. Total IgM and IgG serum levels are significantly lower in GLUT1 deleted B cells [27]. We detected significantly low levels of total IgA, IgG, and IgM without a history of severe infections during infancy and in a later follow-up, and severe hypogammaglobulinemia was diagnosed. Future follow-up is directed toward regular vaccination schedule and avoidance of severe infections.

The current standard of care for GLUT1DS is the ketogenic diet (KD), a high-fat diet that raises the levels of ketone bodies in the blood to make them available to the brain [28,29]. This carbohydrate-restricted diet mimics the metabolic state of fasting. It relies on exogenous fat rather than body fat for ketone production, thus maintaining ketosis without weight loss [1]. As the developing brain requires substantially more energy in young children, KD should be started as early as possible whenever GLUT1DS is suspected and should be continued at least until adolescence [1,10].

Recently, a modified Atkins diet has also been successfully used in patients with GLUT1DS, and it may offer a good alternative in schoolchildren and adolescents with difficulties maintaining a classical KD [10]. Seizure control when managed with a classical KD or MAD is good; one survey of GLUT1DS families described 80% of patients with >90% seizure reduction, including 64% of patients who no longer required antiseizure drugs [7]. Movement disorders and cognitive issues also improve with dietary KD [1]. The efficacy of the ketogenic diet (KD) on cognitive functions has been poorly investigated. Only a few reports have briefly examined the effect of KD on cognitive functioning by standard cognitive tests. Younger patients have demonstrated the most noteworthy response to KD [30]. A recent retrospective study in Italy involving 25 patients (3.7–40 years) with established GLUT1DS diagnosis demonstrated the efficacy of KD on cognitive outcome, confirming that earlier initiation of the diet may prevent the onset of all GLUT1DS symptoms: epilepsy, movement disorders, and cognitive impairment [31]. A retrospective observational study by Amalou et al. reported a positive improvement in seizure reduction, movement disorder, and intellectual functioning and overall better compliance [32]. The efficacy of MAD on cognitive development has rarely been reported in patients with GLUT1DS.

In our patient, we chose MAD as the first-line dietary treatment according to our clinic protocol. We started it early, with suspicion in GLUT1DS and without genetic confirmation. The clinical plan with MAD over classical KD was to closely monitor the patient for 6 months. MAD was started with a carbohydrate intake of 15 grams daily, unrestricted proteins, and an increase in fat amount. Good compliance with the diet plan was reported, ketosis was quickly achieved (3–5 mmol/L), and after 2 weeks, there were no detected seizures and head movements. As we had excellent clinical result in total seizure reduction, good compliance, and no adverse effects, the patient is still on MAD. Improvements in gross and fine motor skills could not be explained with the MAD treatment based on current knowledge; it is more a result of the parents’ everyday activities and exercise. 

Future therapies for GLUT1DS are focusing on supplemental brain metabolic fuels, *SLC2A1* transfer, and small molecules designed to enhance GLUT1 expression or activity [7,28]. Unresolved treatment issues involve the concomitant use of antiseizure drugs, compounds such as acetazolamide, cannabidiol, ketone salts, and ketone esters, and the pharmacological treatment of paroxysmal events, dystonia, and dysarthria, which significantly impair the patient’s quality of life [7]. 

Preclinical experiments on GLUT1DS model mice using AAV9 vectors have demonstrated that gene replacement is effective and durable: brain GLUT1 expression and CSF glucose concentrations increase, brain growth and volume are maintained during development, motor performance is preserved, seizure activity is controlled, and brain angiogenesis proceeds normally [7]. For example, GLUT1 could be delivered to cells by fusing it to a cell-penetrating peptide, such as tat, the protein transduction domain of the human immunodeficiency virus. Preclinical work on model mice could prove instructive in this regard [28].

Triheptanoin is an odd-chain (C7) triglyceride promoted as an alternative to KD that can be metabolized into both acetyl CoA and propionyl CoA—a source of C3 ketone bodies that easily traverse the BBB through MCT1—triheptanoin. In an open-label study, it reduced spike wave discharges in patients. However, a randomized, blinded, placebo-controlled clinical trial to assess the effects of the drug on GLUT1DS patients experiencing seizures or disabling paroxysmal movement disorders failed to demonstrate a benefit, suggesting that triheptanoin is of limited therapeutic use [3,28].

Small molecules may also be used to increase GLUT1 expression and/or its activity. This is especially feasible from a therapeutic perspective, considering the presence of at least one intact *SLC2A1* gene in most GLUT1DS patients, and could be informed by the abundant literature describing molecular pathways that converge onto the GLUT1 protein [28].

In summary for the broad spectrum of clinical phenotypes in GLUT1-associated syndromes, the clinical history should include an assessment of the association of seizures or dyskinesia with fasting states or motor activity and/or improvement after meals. These features are suspicious for GLUT1DS but can also be absent. Therefore, all patients with early-onset absences and generalized epilepsies associated with mental retardation and paroxysmal dyskinesias should undergo a GLUT1 evaluation. The clinical workup includes a detailed medical history and neurological examination, as well as neuropsychological testing, EEG, lumbar puncture, and *SLC2A1* sequencing [2].

## 5. Conclusions

Early suspicion, identification, and molecular confirmation of GLUT1DS are important as an effective dietary treatment is available for a favorable cognitive outcome. Rarely reported, hypogammaglobulinemia in GLUT1DS patients could expand the phenotype of the syndrome outside of neurological dysfunction and the importance of maintaining normal childhood development. The impact of MAD on the cognitive functioning of patients with GLUT1DS is an area of interest, and further studies are necessary to define its efficacy. 

## Data Availability

The data presented in this study are available on request from the corresponding author. The data are not publicly available for privacy reasons.

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
