# Peer review of "GLUT1 Deficiency Syndrome—Early Treatment Maintains Cognitive Development? (Literature Review and Case Report)"

_genes, 2021, doi:10.3390/genes12091379_

Round 1
Reviewer 1 Report
Authors describe a case of GLUT 1 deficiency syndrome with onset in infancy with seizures, which responded to modified Atkin’s diet. Authors claim that the early introduction of diet helped maintain a normal cognitive development. Age of onset and low CSF glucose raised the suspicion for this condition, which was then confirmed by molecular genetic analysis.
Few suggestions below:
- Given that clinical phenotype can vary from normal intellect to severe intellectual disability, is it possible that this could have been a milder disease that did not affect intellectual function, rather than concluding that the introduction of MAD prevented cognitive deficits? Authors have cited literature to express that although deletions are associated with the moderate classical phenotype of GLUT1DS, the genotype-phenotype correlation in GLUT1DS is complex and not clearly defined. Authors should leave the possibility of this being a milder version of disease open if this is an area of ongoing research.
- Also, in discussion section, authors cited literature about the role of Ketogenic diet in improving cognitive function but not about how MAD may address cognitive issues or literature regarding MAD’s role on cognition. Would suggest explicitly stating the rationale behind choosing MAD over KD in this patient.
- Under EEG findings in GLUT1DS, authors state that focal slowing and epileptiform discharges are more prevalent in infants and generalized spike wave pattern in seen in older children. They should clarify what type of epileptiform discharges are seen in infants (multifocal?). Because generalized spike wave discharges also fall under the category of epileptiform discharges.
- Most of the literature suggests low CSF and low lactate as the findings in GLUT1DS. Authors should explain/discuss the normal lactate in this patient.
- Authors cite 1p35-31.3 as the location of SLC2A1 gene, while alternative literature mentions 1p34.2 as the location of the gene. Authors should explain the discrepancy.
- Under conclusion, authors should acknowledge that the role of MAD in cognitive function is an area of ongoing research and further studies are necessary to know the efficacy of the intervention.
Author Response
Than you very much for your review, and comments that will improve our paper.
Point 1. Given that clinical phenotype can vary from normal intellect to severe intellectual disability, is it possible that this could have been a milder disease that did not affect intellectual function, rather than concluding that the introduction of MAD prevented cognitive deficits? Authors have cited literature to express that although deletions are associated with the moderate classical phenotype of GLUT1DS, the genotype-phenotype correlation in GLUT1DS is complex and not clearly defined. Authors should leave the possibility of this being a milder version of disease open if this is an area of ongoing research.
Response 1. Our only explanation is that the patient was diagnosed at early age and course of disease, and maybe we modified GLUT1DS course with early treatment introduciton. We will ephasized the importance of ongoing research need.
Point 2. Also, in discussion section, authors cited literature about the role of Ketogenic diet in improving cognitive function but not about how MAD may address cognitive issues or literature regarding MAD’s role on cognition. Would suggest explicitly stating the rationale behind choosing MAD over KD in this patient.
Response 2. We choose MAD over KD because it is standard at our Clinic, with closer check-up of patient. As results were good in symptoms reduction, compliance at absence of adverse effect we continued it.
Point 3. Under EEG findings in GLUT1DS, authors state that focal slowing and epileptiform discharges are more prevalent in infants and generalized spike wave pattern in seen in older children. They should clarify what type of epileptiform discharges are seen in infants (multifocal?). Because generalized spike wave discharges also fall under the category of epileptiform discharges.
Response 3. We will correct it better in text.
Point 4. Most of the literature suggests low CSF and low lactate as the findings in GLUT1DS. Authors should explain/discuss the normal lactate in this patient.
Response 4. According to our laboratory reference values it is normal value (0,9 mmol/l; ref.values 0,8-2,0mmol/l), but towards low limits value. So that is the only explanation. Some literature states that lactate are low to normal in patients with GLUT1DS (Klepper, J.; Akman, C.; Armeno, M.; Auvin, S.; Cervenka, M.; Cross, H. J.; De Giorgis, V.; Della Marina, A.; Engelstad, K.; Heussinger, N.; Kossoff, E. H.; Leen, W. G.; Leiendecker, B.; Monani, U. R.; Oguni, H.; Neal, E.; Pascual, J. M.; Pearson, T. S.; Pons, R.; Scheffer, I. E.; Veggiotti, P.; Willemsen, M.; Zuberi, S. M.; De Vivo, D. C. Glut1 Deficiency Syndrome (Glut1DS): State of the Art in 2020 and Recommendations of the International Glut1DS Study Group. Epilepsia Open 2020, 5 (3), 354–365. )
Point 5. Authors cite 1p35-31.3 as the location of SLC2A1 gene, while alternative literature mentions 1p34.2 as the location of the gene. Authors should explain the discrepancy.
Response 5. Some articles cite this 1p35-31.3 location, and some 1p34.1. We could not explain the discrepancy, so decided to change cytogenetic location in article according to OMIM
Point 6. Under conclusion, authors should acknowledge that the role of MAD in cognitive function is an area of ongoing research and further studies are necessary to know the efficacy of the intervention.
Respoense 6. We will absolutelly do that
Reviewer 2 Report
See Attached file

Author Response
Than you very much for your opinions and comments, and we will do our best to maximally improve our article.
Point 1. This article does not provide new knowledge to the understanding of GLUT1-DS. It is important to make this condition known among colleagues, but the typical clinical picture and the efficacy of ketogenic diet in GLUT1 DS is already well known.
Response 1. We wanted to emphasize need for early recognition and treatment of this patients in setting of cgood clinical practice.
Point 2. The title indicates a case report and a literature review of this rare condition, but the paragraph Material and Methods does not include any information on how the review was performed. Hence, the literature search seems coincidentally. In the case in question, information about why the different investigations, e.g. immunological investigations, was performed is lacking. Moreover, the technique for genetic analyses is unnecessary deeply described.
Response 2. We will shorten the genetic technique.
As standard routine testing of spinal fluid of young patients with epilepsy in our Clinic consists of BBB function test (and need for serum IgG test), and that is how we accidentally found hypogammaglobulinemia.
Point 3. Under the paragraph Results, the case report lack information of how the developmental level of the child is evaluated. Information of the specific mutation/deletion found in this case is also missing. Why did they choose to start with modified Atkins diet when classical ketogenic diet is the first treatment choice in children with newly diagnosed GLUT1-DS?
Response 3. We corrected it, and explained in more detailes developmental evaluation of patient.
We provided information of identification full coding sequence deletion of the SLC2A1 gene.
Introduction of MAD is standard protocol in our Clinic, and we decided to start with it with close check-up of our patient for 6 months. As the clinical improvement of total seizures reduction, excellent compliance and no adversede effects was noticed we are still continuing with it.
Point 4. Under the paragraph Discussion, the authors categorize the patient as a classic phenotype of GLUT1-DS with myoclonic seizures and head nodding starting in infancy. That may be correct. Nevertheless, a discussion of how the treatment may affect the phenotype is missing. If untreated, the classic GLUT1 DS patient develops epilepsy, microcephaly, developmental delay, and quite often paroxysmal or permanent movement disorders. It is not known how the development of this specific child would have been without dietary treatment.
Response 4. It is true, dietary treatment modified disease course.
Point 5. The authors also state that the majority of patients carry de novo variants, and that only rare families with SLC2A1 mutations are described with an autosomal dominant mode of inheritance. It is probably correct that the majority of patients have a de novo mutation. However, in 2017 we published an article titled “GLUT1-deficiency syndrome; Report of a four-generation family with a mild phenotype”. Here we describe a family with 10 affected members. Most of them had normal IQ, although some had learning difficulties.
Response 5. We have read this paper, and it is excellent, but we just provided short information on genetic basis of disease. as our patients carries de novo mutation.
Point 6. It seems that English is not the authors’ first language. There are many spelling errors, and the language needs to be improved.
Response 6. We did our best to improve it
Point 7. In my opinion, this article brings no new knowledge to the field of GLUT1-DS.
Response 7. We agree that there is not much new knowledge in filed of GLUT1DS, but wanted to emphasize the importance of early diagnosis, genetic confirmation and introduciton of treatment, with importance of further research needed in this field
Reviewer 3 Report
Thank you for submitting this manuscript, "GLUT1 Deficiency Syndrome – Early treatment maintains cognitive development? (Literature review and case report)."
- The case is interesting and advocates a fundamental concept that early diagnosis and early treatment with a ketogenic diet should be considered for the best neurocognitive outcome, but there are many unusual findings like the presence of early motor delays improving with diet, initiating treatment with modified Atkins diet instead of the ketogenic diet, presence of hypo-gammaglobulinemia, large deletion mutations which are usually associated with a significant cognitive deficit, challenging to evaluate neurocognitive profile at this younger age.
- Traditionally, the ketogenic diet is considered being superior to Modified Atkins. Lately, more data has come up which have stated that they are more comparable in terms of seizures outcome. However, data dealing with other parameters are still lacking, especially in younger age groups. It is unusual and not a common practice to use a modified Atkins diet, especially in infants, as it is reserved mainly in older children and adolescents. Can the authors explain why they did not choose a Ketogenic diet in this infant?
- Since the theme of the case is maintaining good cognitive development with a Ketogenic diet in the setting of preexisting motor delay and significant deletion mutation; the child should preferably have an objective evaluation for neurocognitive parameters, which are considered age-appropriate for a 2-year-old, which should be highlighted in the case report.
- Can the author elaborate on the association of hypogammaglobulinemia with glut-1 deficiency and reported cases of its association?
- How does the author explain much benign course with deletion mutation?
- Infants with GLUT-1 deficiency usually have social and language dysfunction, and it is unusual to have isolated motor deficit, especially in the early course of the disease. The case findings are quite contrary (mild motor delay and normal cognition), but I agree with the author's report that early ketogenic diet initiation ensures the best cognitive outcome. It is unusual that motor skills have normalized with the Ketogenic diet. The author should elaborate on the mild gross motor impairment in the case report.
Author Response
Thank you for this review and comments for improvement of our paper.
Point 1. Traditionally, the ketogenic diet is considered being superior to Modified Atkins. Lately, more data has come up which have stated that they are more comparable in terms of seizures outcome. However, data dealing with other parameters are still lacking, especially in younger age groups. It is unusual and not a common practice to use a modified Atkins diet, especially in infants, as it is reserved mainly in older children and adolescents. Can the authors explain why they did not choose a Ketogenic diet in this infant?
Response 1. Standard protocol at our Clinic is introduction of Modified Atkins diet, and we decided to start it for six months with close check-up of patient. As we observed good clinical response in total reduction of seizures, and normal development without any adverse effects we decided to continue with it. Ofcourse, if we in any time see no improvement or deteriorations we would transfer our patient to classic ketogenic diet.
Point 2. Since the theme of the case is maintaining good cognitive development with a Ketogenic diet in the setting of preexisting motor delay and significant deletion mutation; the child should preferably have an objective evaluation for neurocognitive parameters, which are considered age-appropriate for a 2-year-old, which should be highlighted in the case report.
Response 2. We performed objective neuropsychological evaluation in our patient. We corrected it in text.
Point 3. Can the author elaborate on the association of hypogammaglobulinemia with glut-1 deficiency and reported cases of its association?
Response 3. We found hypogammaglobulinemia in our patient accidentally (when we perform lumbar puncture in young patients with epilepsy we test function of BBB and then found low levels of IgG first and IgA and IgM), and tried to research its association. Our only explanation is defective glut1 transporter in B lymph, but did not find any available test or evidence to confirm it.
Point 4. How does the author explain much benign course with deletion mutation?
Response 4. We can only explain it with early diagnose and introduction of treatment, and possible change of GLUT1DS disease course. If unrecognised, maybe course would not be so benign.
Point 5. Infants with GLUT-1 deficiency usually have social and language dysfunction, and it is unusual to have isolated motor deficit, especially in the early course of the disease. The case findings are quite contrary (mild motor delay and normal cognition), but I agree with the author's report that early ketogenic diet initiation ensures the best cognitive outcome. It is unusual that motor skills have normalized with the Ketogenic diet. The author should elaborate on the mild gross motor impairment in the case report.
Response 5. It could be the age of patient, were there could be some transitional changes in motor functioning during examination, but we noticed them. We don't think that it is strictly connected with diet. We elaborated details of neuropsychological tests in text.
Round 2
Reviewer 3 Report
Thank you for submitting this revised manuscript. The authors have done a nice job in addressing the reviewer's comments. The manuscript has significantly improved and is now acceptable for publication, but the authors should consider elaborating on the Modified Atkins Diet protocol at their center for infants and toddlers in Methods section.
Author Response
Thank you very much for suggestions. We added our Clinic's MAD Protocol in Section Methods.